# MODALITY-AWARE QUANTIZATION: BALANCING VISUAL AND TEXTUAL FIDELITY IN MULTIMODAL COMPRESSION

## ABSTRACT

Vision-language models (VLMs) have achieved remarkable capabilities across multimodal tasks, yet their deployment remains constrained by substantial computational requirements. While post-training quantization (PTQ) offers a practical solution for model compression, existing methods fail to address a fundamental challenge in VLM quantization: the inherent heterogeneity between visual and textual representations. In this work, we identify and formalize a critical failure mode where visual tokens, despite their lower semantic density, dominate the quantization optimization process due to their extreme value distributions and numerical prevalence. This dominance systematically degrades the preservation of semantically-critical language tokens, severely impacting model performance. We present a theoretically grounded framework that proves this trade-off through formal analysis and introduce an adaptive optimization pipeline that dynamically balances cross-modal heterogeneity. Our method leverages activation-scale statistics and gradient-sensitivity priors to construct layer-wise modality weights that counteract visual dominance while preserving linguistic fidelity—all without altering the inference computation graph. Extensive experiments demonstrate state-of-the-art performance across diverse quantization regimes: on Qwen-VL-Chat with W4A8 quantization, we achieve $59.27\%$ on TextVQA, surpassing the previous best method MQuant by $+2.70\%$. Most notably, under extreme W4A4 quantization where existing approaches fail catastrophically, our method maintains robust performance ($55.72\%$ on TextVQA), proving that aggressive multimodal compression is both achievable and practical for real-world deployment. The code is available at this anonymous link.

## 1 INTRODUCTION

Large language models (LLMs) have achieved remarkable progress in language understanding, reasoning, and generation tasks (Vaswani et al., 2017; Brown et al., 2020; Hoffmann et al., 2022). However, the substantial increase in model parameters and context length has led to significant deployment challenges, including excessive memory consumption, high activation transfer bandwidth requirements, and increased end-to-end latency (Dettmers et al., 2022; Dao et al., 2022; Press et al., 2021). These computational demands limit the practical deployment of LLMs in resource-constrained environments. Consequently, model compression and acceleration techniques have become critical for enabling efficient LLM deployment under strict latency and memory constraints, transforming these models from research prototypes into practical applications.

Among existing compression techniques, post-training quantization (PTQ) has emerged as a particularly attractive approach due to its minimal data requirements and computational efficiency. Unlike other methods that necessitate extensive fine-tuning or modifications to the pretraining pipeline, PTQ requires only a small calibration dataset to achieve effective compression. Recent advances in LLM quantization have yielded significant improvements through three key innovations: refined joint optimization strategies for weight and activation quantization (Sun et al., 2024b; Ma et al., 2024; Shao et al., 2023; Liu et al., 2024b), enhanced calibration objectives that better preserve model performance (Frantar et al., 2022; Ashkboos et al., 2024; Li et al., 2025b), and sophisticated

magnitude-management techniques (Xiao et al., 2023; Lin et al., 2024; Ma et al., 2024; Ashkboos et al., 2024; Liu et al., 2024b) that address outlier values and dynamic range challenges.

Concurrently, the proliferation of multimodal data and applications has driven the development of vision-language models (VLMs) that demonstrate robust generalization and reasoning capabilities across diverse tasks, including image captioning, visual question answering (VQA), document understanding, and multi-turn image-text dialogue (Alayrac et al., 2022; Liu et al., 2023). Contemporary VLM architectures typically consist of three core components: a vision encoder for extracting visual features, a modality projector or alignment layer that maps these features into the language embedding space, and a large language decoder for generating textual outputs (Radford et al., 2021; Dai et al., 2023; Liu et al., 2024a). These models achieve cross-modal alignment and conversational competence through a two-stage training paradigm: large-scale pretraining on image-text pairs followed by task-specific instruction tuning.

Despite these advances, research on VLM quantization remains limited and fragmented compared to the extensive work on LLM compression. Current approaches predominantly adapt LLM-oriented PTQ techniques to multimodal settings without accounting for the unique characteristics of visual components, often restricting evaluation to the language decoder alone (Wang et al., 2024a). Notably, comprehensive quantization strategies that explicitly model both the vision encoder and projection layer remain unexplored. VLM quantization presents distinct challenges arising from the inherent heterogeneity between visual and textual representations, which differ substantially in their magnitude distributions, outlier patterns, and sensitivity to perturbations. While preliminary solutions incorporating modality-specific scaling factors and sensitivity-based weighting schemes have shown promise, the field lacks a unified end-to-end quantization framework that encompasses all VLM components while preserving the original inference computation graph.

A detailed analysis of the VLM computational pipeline reveals the fundamental sources of these quantization challenges. Vision encoders produce activations characterized by significantly higher variance and heavier-tailed distributions compared to their language-only counterparts, with visual token sequence lengths varying according to image resolution and encoder architecture. The projection layer, which facilitates cross-modal integration through learned linear transformations and normalization operations, exhibits particular vulnerability to quantization-induced perturbations that can disrupt its carefully calibrated invariance properties. Within the language decoder, model performance shows disproportionate sensitivity to quantization errors in specific attention and feed-forward blocks; consequently, uniform calibration strategies risk biasing quantization parameters toward visual statistics at the expense of linguistic precision. From a systems perspective, practical deployment constraints—including GPU memory limitations, bandwidth requirements for vision-to-decoder communication, and KV-cache scaling during extended context generation—compound these challenges, amplifying the impact of suboptimal quantization decisions on overall system performance. Based on the above observations, we formalize the calibration objective for VLMs and establish a trade-off property: overemphasizing the reconstruction of visual tokens necessarily degrades the fidelity of language tokens, whereas down-weighting the visual term tightens the upper bound of language-side error. Empirically, token-wise statistics on Qwen-VL reveal a two-order-of-magnitude disparity in activation scales across modalities, and ablations further show that, under identical quantization budgets, textual tokens are substantially more sensitive to perturbations.

Accordingly, in optimizing the language branch of VLMs, we introduce a forward-based, modality-weighted reconstruction objective that adaptively rebalances visual and textual errors by leveraging activation-scale ratios and a gradient-gap prior, thereby steering calibration to more effectively reduce language-token error.

Our contributions are summarized as follows:

- We identify and formalize visual token dominance in VLM quantization. Despite carrying less semantic information, visual tokens overwhelm optimization due to their extreme distributions and numerical abundance, systematically degrading language token precision—a critical failure mode we prove through Theorem 1.

- We introduce an adaptive optimization pipeline that balances cross-modal heterogeneity. Our method combines activation statistics with gradient priors to construct dynamic modality weights, counteracting visual dominance while preserving linguistic fidelity.

- We achieve state-of-the-art performance across all quantization regimes. On Qwen-VL-Chat (W4A8), we surpass MQuant by $+2.70\%$ on TextVQA/MME (59.27% vs. 56.57%). Under extreme W4A4 quantization where MQuant fails catastrophically (0/28), our method maintains strong performance (55.72%/1322), proving aggressive multimodal compression is practical.

## 2 RELATED WORK

**General PTQ for LLMs.** Research on post-training quantization (PTQ) for large language models has progressed rapidly, establishing strong baselines and a clear design space. Weight-only methods such as GPTQ(Frantar et al. (2022)) leverage approximate second-order information to perform one-shot rounding with minimal calibration cost, often reaching 3–4-bit precision with negligible loss. Activation-aware techniques including SmoothQuant(Xiao et al. (2023)) mitigate outliers by migrating activation difficulty into weights via equivalent reparameterizations, stabilizing joint weight–activation quantization without altering the inference graph. Channel-selective and hardware-oriented approaches like AWQ(Lin et al. (2024)) prioritize a small set of critical channels to enable INT3/INT4 deployment, while global calibration schemes such as OmniQuant(Shao et al. (2023)) and AffineQuant(Ma et al. (2024)) align reconstruction objectives and learn affine transforms to reduce accumulated error across layers. Rotation-based, computation-invariant methods (e.g., QuaRot(Ashkboos et al. (2024)), SpinQuant(Liu et al. (2024b)), FlatQuant(Sun et al. (2024b))) further flatten heavy-tailed activations and decorrelate channels through orthogonal or learnable rotations, enabling robust 4-bit inference for weights, activations, and even KV caches under practical serving constraints.

**Vision–Language Models (VLMs).** In parallel, the surge of multimodal applications has driven the evolution of VLM architectures that integrate a vision encoder, a projector/alignment module, and a large language decoder. Systems such as Flamingo demonstrate few-shot generalization on interleaved image–text sequences by bridging strong vision backbones with language models; BLIP-2 attains efficient alignment with a lightweight query transformer that allows frozen backbones; and LLaVA operationalizes visual instruction tuning to form end-to-end conversational systems. Engineering-scale families including Qwen-VL/Qwen2-VL and InternVL2 extend these designs with dynamic resolution handling, improved modality alignment, and long-context inference, thereby providing widely used platforms for downstream quantization research. Across these models, practical deployment hinges on preserving cross-modal alignment and decoding quality while managing memory footprint, bandwidth between stacks, and latency.

**PTQ tailored to VLMs.** As vision–language models (VLMs) continue to scale, generic LLM quantization techniques show clear limitations in multimodal scenarios, leading to a series of specialized approaches. Q-VLM(Wang et al. (2024a)) was the first to systematically formulate post-training quantization for full VLM stacks, analyzing error propagation across the multimodal pipeline and introducing end-to-end calibration strategies that preserve the inference graph while ensuring stable compression, thus establishing a strong baseline for subsequent research. P4Q(Sun et al. (2024a)) extends this line by introducing a prompt-based mechanism during calibration, learning task-adaptive prompts that guide the quantizer to exploit multimodal context, thereby achieving stronger robustness under aggressive low-bit settings. MBQ(Li et al. (2025a)) highlights the pronounced sensitivity gap between language and visual tokens and proposes gradient-driven modality weighting during calibration, which significantly improves low-bit accuracy without altering the inference computation graph. VLMQ(Xue et al. (2025)) argues that conventional Hessian-based PTQ treats all tokens as exchangeable, allowing redundant visual tokens to bias second-order statistics and yield suboptimal weight rounding. To address this, VLMQ proposes importance-weighted second-order calibration: a single block-local backpropagation pass produces gradient signals used to compute token-level importance factors. These factors parameterize a weighted Hessian/objective that guides rounding so that weight perturbations align with token importance. Relative to sensitivity-only schemes, this approach anchors the calibration objective in second-order structure while retaining efficiency via local, one-pass estimation. The result is a principled mechanism that reduces over-fitting of quantizers to visually redundant structure and preserves task-relevant linguistic cues. MQuant(Yu et al. (2025)) highlights pronounced scale disparities between visual and textual tokens. Using a single global scale either enlarges the dynamic range to accommodate visual

tails—coarsening quantization for text—or tightens the range for text at the cost of frequent clipping for vision. MQuant therefore introduces modality-specific static quantization (MSQ), which learns and fixes per-channel-group scaling parameters separately for visual and textual tokens. This captures modality differences while avoiding per-token dynamic scaling, striking a favorable balance between low latency and high accuracy. In practice, MQuant makes explicit that the choice of scaling granularity is central to VLM quantization: aligning scale to modality boundaries reduces tail-driven outliers and improves utilization of the available bit budget.

# 3 METHODOLOGY

## 3.1 PRELIMINARY

**Post-Training Quantization**. Quantization reduces model size and computational overhead by mapping high-precision floating-point representations (e.g., BF16, FP16, FP32) of weights and activations to low-precision fixed-point integers (e.g., INT8, INT4) or low-precision floating-point formats (e.g., FP8, FP4). Specifically, the application of asymmetric fixed-point quantization to a given tensor $X$ can be formalized as follows:

$$Q(X) = clamp\left(\left\lfloor \frac{X}{s} \right\rceil + zp, 0, 2^n - 1\right), s = \frac{\max(X) - \min(X)}{2^n - 1}, \quad (1)$$

where $clamp(\cdot)$ clips the quantized values to the quantization range. $zp$ denotes the zero point, $n$ represents the bit width, and $s$ is the quantization scale factor (or step size). For a linear layer module with weights $W$ and activations $X$, the quantized matrix operations are expressed as follows:

$$Y = \hat{X}\hat{W} = s_x \cdot s_w \cdot (Q(X) - zp_x)((Q(W) - zp_w). \quad (2)$$

Where $s_x$ and $s_w$ denote the quantization scale factors for the activation tensor $X$ and weight tensor $W$, respectively, while $zp_x$ and $zp_w$ represent their corresponding zero points. To assess quantization error in models, the L2 norm of the feature difference between pre- and post-quantization states serves as a standard evaluation metric. Formally, this measure is expressed as:

$$\mathcal{L}(W, X) = \|f(W, X) - f(\hat{W}, \hat{X})\|_2^2. \quad (3)$$

Where $f(\cdot)$ represents the model function. In the specific case where $f$ corresponds to a single linear layer, Equation 3 simplifies to the expression $\|WX - \hat{W}\hat{X}\|_2^2$, representing the difference between the original computation and its quantized counterpart. Existing approaches (Xiao et al., 2023; Shao et al., 2023; Ma et al., 2024; Ashkboos et al., 2024; Liu et al., 2024b; Sun et al., 2024b) mitigate quantization error by introducing additional parameters that redistribute outliers in weight and activation tensors. Denoting these additional parameters as $\Theta$, the modified quantization error loss function is formulated as:

$$\mathcal{L}(W, X, \Theta) = \|f(W, X) - f(\hat{W}, \hat{X}, \Theta)\|_2^2. \quad (4)$$

**Visual Language Model.** Vision-language models (VLMs) (Wang et al., 2024b; Bai et al., 2025; Chen et al., 2024; Wang et al., 2024c; Hu et al., 2024; Yao et al., 2024) process multimodal inputs simultaneously, enabling diverse applications across domains. VLM architectures typically consist of three core components: a vision encoder, a language model, and a connector module. The vision encoder (Dosovitskiy et al., 2020; Radford et al., 2021; Ilharco et al., 2021) transforms raw visual inputs into latent feature representations. Current state-of-the-art VLMs predominantly utilize Vision Transformer architectures as vision encoders due to their superior feature extraction and compression capabilities. The connector module bridges the dimensional gap between vision encoder outputs and language feature spaces. The language model processes both visual tokens and textual embeddings concurrently, concatenating these multimodal representations during forward propagation to produce unified outputs. Given this multimodal processing pipeline, Equation 4's quantization error formulation must be refined to account for the distinct characteristics of visual and textual tokens:

$$\mathcal{L}(W, X, \Theta) = \|f(W, X_l) - f(\hat{W}, \hat{X}_l, \Theta)\|_2^2 + \|f(W, X_v) - f(\hat{W}, \hat{X}_v, \Theta)\|_2^2. \quad (5)$$

where $X = \{X_l, X_v\}$, with $X_l$ and $X_v$ representing the language and visual tokens, respectively.

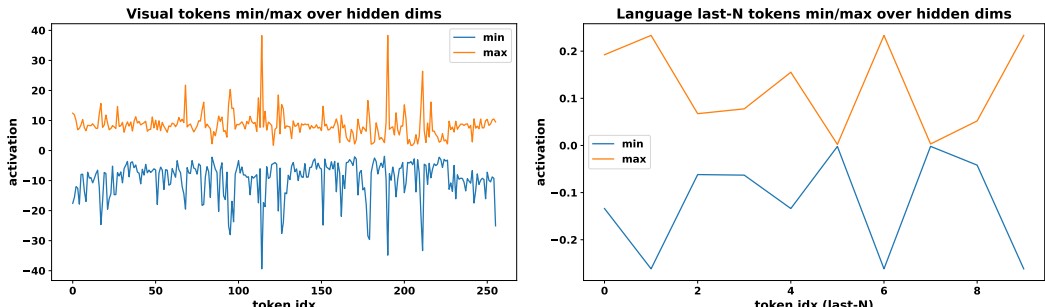

Figure 1: Token-wise activation ranges for visual vs. textual tokens. Curves report per-token minimum/maximum over hidden dimensions. Visual tokens exhibit heavy-tailed, large-range activations with outliers up to ±40, whereas textual tokens remain confined within roughly ±0.25. The two-order-of-magnitude gap motivates modality-aware calibration/weighting for VLM quantization.

## 3.2 Optimization Analysis

Unlike large language models, vision-language models process visual and textual inputs simultaneously, necessitating quantization schemes that accommodate the heterogeneous distributions of multimodal features, as formalized in Equation 5. Figure 1 presents the distribution of visual and textual tokens within the same feature layer of the QwenVL model, revealing two critical observations. First, visual tokens exhibit significantly more extreme distributions than textual tokens. The absolute maximum values of visual tokens reach approximately 40, whereas textual tokens peak at only 0.2—a two-order-of-magnitude disparity. This substantial difference causes visual token quantization errors to dominate the loss function in Equation 5, leading parameter optimization $\Theta$ to prioritize visual token distribution fitting at the expense of textual token accuracy. Second, visual tokens substantially outnumber textual tokens in the language model. This imbalance arises from the vision encoder's patch-based decomposition of raw images into multiple visual tokens. However, these visual patches possess low information density (Gholami et al., 2025); background patches, for instance, can be eliminated without degrading model performance. Consequently, although visual tokens constitute a large proportion of the input, their inherent information redundancy renders them relatively insensitive to quantization-induced compression (Li et al., 2025a).

Building upon these observations, we initially formulate the weighted quantization optimization process as follows:

$$\Theta^* = \arg\min_{\Theta} \|f(W, X_l) - f(\hat{W}, \hat{X}_l, \Theta)\|_2^2 + \alpha\|f(W, X_v) - f(\hat{W}, \hat{X}_v, \Theta)\|_2^2. \tag{6}$$

For clarity in the subsequent proof, we adopt simplified notation. We denote the quantization error loss of the language component as $L_l(\Theta) = \|f(W, X_l) - f(\hat{W}, \hat{X}_l, \Theta)\|_2^2$ and that of the visual component as $L_v(\Theta) = \|f(W, X_v) - f(\hat{W}, \hat{X}_v, \Theta)\|_2^2$. Consequently, the total weighted quantization loss function can be expressed as: $\mathcal{L}(\Theta, \alpha) = L_l(\Theta) + \alpha L_v(\Theta)$. We then present the following theorem to establish how visual token quantization error affects language token performance.

**Theorem 1 (Principle of Objective De-emphasis)** *Let $L_A(\Theta)$ and $L_B(\Theta)$ be two non-negative loss functions parameterized by $\Theta \in \mathbb{R}^d$. Consider the composite objective function $J(\Theta, \alpha) = L_A(\Theta) + \alpha L_B(\Theta)$, where $\alpha \geq 0$ is a scalar weight.*

*Let $\alpha_1$ and $\alpha_2$ be two weights such that $\alpha_1 > \alpha_2 \geq 0$. Let $\Theta_1 = \arg\min_{\Theta} J(\Theta, \alpha_1)$ and $\Theta_2 = \arg\min_{\Theta} J(\Theta, \alpha_2)$, assuming such minimizers exist.*

*Then the following trade-off relationship holds:*

$$L_A(\Theta_2) \leq L_A(\Theta_1), \tag{7}$$
$$L_B(\Theta_1) \leq L_B(\Theta_2). \tag{8}$$

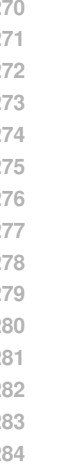
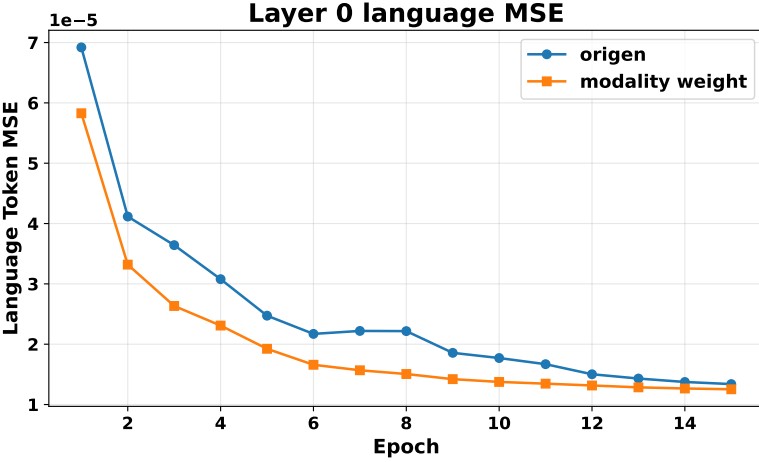

Figure 2: Language-token reconstruction error (MSE) over calibration epochs: modality weighting accelerates and lowers language-side error, indicating better focus on textual fidelity by suppressing visual activation dominance.

The proof appears in Appendix A.2. This theorem demonstrates that visual loss dominance ($\alpha_1 > \alpha_2$) yields parameters $\Theta_1$ that compromise textual loss optimization, resulting in $L_A(\Theta_2) < L_A(\Theta_1)$. Figure 2 illustrates the relationship between language loss and varying visual token MSE weights. Reducing visual loss weighting produces superior textual loss convergence compared to the original loss distribution. Experiments confirm this balance critically affects vision-language model performance. The high information density of language tokens makes them particularly vulnerable to lossy compression—even minimal information degradation substantially reduces accuracy on multimodal benchmarks. As a single word modification can fundamentally alter the semantic meaning. The following section presents a detailed framework for optimizing the trade-off between these competing loss components.

### 3.3 ADAPTIVE OPTIMIZATION PIPELINE

**Pipeline.** Under the joint requirements of **modality heterogeneity** and **computation invariance**, our method proceeds in three stages. **(i) Vision-side normalization replacement.** We replace the normalization operators in the vision stack with **RMSNorm**, whose orthogonal invariance and positive homogeneity allow calibration-time rotations and magnitude redistributions to be folded into parameters, keeping the vision branch and the projector computation-invariant. **(ii) FlatQuant encapsulation.** We adopt *FlatQuant* as the quantization engine and apply its rotation–reconstruction routine **end to end** across the vision encoder, the modality projector/connector, and the language decoder. **(iii) Forward-based modality weighting (language branch only).** During calibration, we use forward-pass statistics to characterize cross-modal activation properties and combine them with the observed gradient disparity between textual and visual tokens to form layer-wise modality coefficients, which weight reconstruction errors of visual versus textual tokens in the language stack; the vision stack uses the standard unweighted MSE.

**Notation.** Let layer $\ell$ in the language stack observe visual/textual token index sets $\mathcal{V}^{(\ell)}$ and $\mathcal{T}^{(\ell)}$, respectively. Denote the pre-quantization output by $Y^{\text{fp}} \in \mathbb{R}^{(\cdot) \times d}$ and the post-quantization output by $Y^{\text{q}} \in \mathbb{R}^{(\cdot) \times d}$. We use the per-token **dimension-normalized $\ell_2$ magnitude** as a scale surrogate,

$$\phi(x) = \frac{\|x\|_2}{d}, \qquad x \in \mathbb{R}^d. \tag{9}$$

**Activation-scale ratio (first coefficient).** Define layer-wise average scales for visual and textual tokens,

$$\bar{S}_{\text{vis}}^{(\ell)} = \frac{1}{|\mathcal{V}^{(\ell)}|} \sum_{t \in \mathcal{V}^{(\ell)}} \phi(Y_{t,:}^{\text{fp}}), \qquad \bar{S}_{\text{txt}}^{(\ell)} = \frac{1}{|\mathcal{T}^{(\ell)}|} \sum_{t \in \mathcal{T}^{(\ell)}} \phi(Y_{t,:}^{\text{fp}}), \tag{10}$$

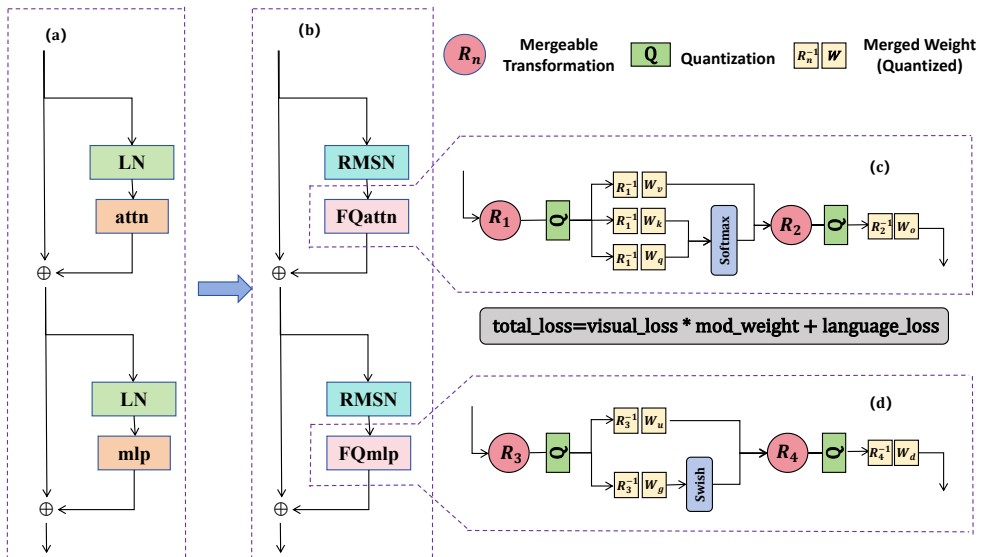

Figure 3: Overview of the quantization pipeline: (a) original Transformer block; (b) replace Layer-Norm with RMSNorm in the vision branch to preserve orthogonality and scale invariance; (c)(d) encapsulate FlatQuant into the attention and MLP modules, fold calibration-time rotations into weights, and deploy the merged quantized weights at inference; additionally, the language branch uses a modality-weighted reconstruction objective, where the total loss combines visual loss and language loss with adaptive weighting to suppress visual activation dominance and better reduce textual-token error.

and the corresponding activation-scale ratio

$$\alpha^{(\ell)} \;=\; \frac{\bar{S}_{\text{txt}}^{(\ell)}}{\bar{S}_{\text{vis}}^{(\ell)}}. \tag{11}$$

**Gradient-disparity prior (second coefficient).** Motivated by Theorem 1 and Figure 2 that textual tokens exhibit markedly larger training-time/sensitivity gradients than visual tokens, we introduce a gradient-gap parameter

$$\beta \;\in\; (0,1], \qquad \text{with a default setting } \beta = 0.1. \tag{12}$$

**Layer-wise modality weights and objectives.** The visual-token weight in the language stack is formed by combining the scale and gradient cues,

$$\lambda_{\text{vis}}^{(\ell)} \;=\; \alpha^{(\ell)} \cdot \beta, \qquad \lambda_{\text{txt}}^{(\ell)} \;=\; 1, \tag{13}$$

leading to the modality-weighted reconstruction loss for layer $\ell$ in the language branch:

$$\mathcal{L}_{\text{lang}}^{(\ell)} = \frac{1}{|\mathcal{V}^{(\ell)}|} \sum_{t \in \mathcal{V}^{(\ell)}} \lambda_{\text{vis}}^{(\ell)} \|Y_{t,:}^{\text{q}} - Y_{t,:}^{\text{fp}}\|_2^2 + \frac{1}{|\mathcal{T}^{(\ell)}|} \sum_{t \in \mathcal{T}^{(\ell)}} \lambda_{\text{txt}}^{(\ell)} \|Y_{t,:}^{\text{q}} - Y_{t,:}^{\text{fp}}\|_2^2. \tag{14}$$

For layers in the vision stack (which only process visual tokens), we retain the standard unweighted objective:

$$\mathcal{L}_{\text{vis}}^{(\ell)} = \frac{1}{N^{(\ell)}} \sum_{t=1}^{N^{(\ell)}} \|Y_{t,:}^{\text{q}} - Y_{t,:}^{\text{fp}}\|_2^2, \qquad \ell \in \mathcal{L}_{\text{vis}}. \tag{15}$$

**Discussion.** This construction unifies rotation-based quantization with modality-aware weighting under a computation-invariant implementation: $\alpha^{(\ell)}$ captures the systematic activation-scale gap

between modalities, $\beta$ encodes the gradient-sensitivity prior for text, and their product $\lambda_{\text{vis}}^{(\ell)}$ suppresses loss dominance from large-range visual activations while maintaining decoder-critical textual fidelity. Because rotations are absorbed into weights and magnitude adjustments are folded into normalization-affine parameters, the inference graph remains unchanged.

## 4 EXPERIMENTS

**Implementation Details.** We conduct all experiments under a consistent software–hardware environment and follow common practice for VLM post-training quantization. In multimodal calibration scenarios, we randomly sample **128** images from the **COCO** dataset to build the calibration set. We also bound the visual token budget to **256** per sample when processing datasets, thereby avoiding additional search overhead when applying the modality-weighted MSE to visual tokens. We apply **per-channel** quantization to weights and **per-token** quantization to activations, thereby avoiding coarse per-tensor scaling; this granularity mitigates the impact of modality heterogeneity (visual vs. textual tokens) on the learned quantization ranges. Unless otherwise noted, decoding hyperparameters, random seeds, and preprocessing pipelines are kept fixed across methods to ensure fair comparisons and reproducibility.

**Baselines.** Our experiments are conducted on **Qwen-VL-Chat** and **InternVL2-8B**. All runs are executed on a single **NVIDIA A800 (80 GB)** GPU, demonstrating favorable efficiency and scalability. We compare against representative baselines, including **RTN** (round-to-nearest), **GPTQ**, and **MQuant** (a strong whole-model VLM quantizer). In terms of bit-width configurations, we evaluate (i) **W8A8 for the vision stack + W4A8 for the language stack** (the MQuant setting), (ii) **W4A8** for both vision and language, and (iii) settings with **4-bit activations** to stress low-bit regimes.

**Evaluation.** To assess the effectiveness of the proposed VLM quantization method, we evaluate on seven multimodal benchmarks: **TextVQA_VAL**, **MME**, **OCRBench**, **DocVQA**, **ScienceQA**, **MMMU**, and **SEEDBench_IMG**. Following the MQuant evaluation protocol, we use the **same version of VLMEvalKit** to compute task-specific metrics (e.g., accuracy or task-defined scores) and the overall average. This comprehensive suite measures the impact of quantization on multiple facets of multimodal understanding and examines generalization across diverse tasks.

Table 1: Evaluation Results of the Quantized Qwen-VL Model across Datasets

| Method | Config | | T'VQA | MME | OCR' | D'VQA | S'QA | MMMU | SEED' |
| | vision | lang | | | | | | | |
|---|---|---|---|---|---|---|---|---|---|
| – | bf16 | bf16 | 60.62 | 1433 | 488 | 58.02 | 62.80 | 33.5 | 63.61 |
| RTN | | | 3.64 | 380 | 149 | 4.59 | 17.60 | 9.1 | 20.84 |
| GPTQ | W8A8 | W4A8 | 17.95 | 463 | 148 | 8.91 | 24.03 | 14.7 | 26.11 |
| MQuant | | | 59.53 | 1286 | 468 | 55.81 | 58.32 | 30.8 | 61.73 |
| Ours | | | 60.18 | 1379 | 471 | 56.92 | 61.61 | 32.2 | 64.08 |
| RTN | | | 1.55 | 368 | 118 | 1.15 | 15.55 | 7.8 | 19.84 |
| GPTQ | W4A8 | W4A8 | 16.54 | 401 | 152 | 6.85 | 24.27 | 12.7 | 22.66 |
| MQuant | | | 56.57 | 1315 | 460 | 53.05 | 60.08 | 29.9 | 61.10 |
| Ours | | | 59.27 | 1374 | 468 | 54.90 | 62.47 | 33.0 | 64.04 |
| RTN | | | 0.35 | 0 | 0 | - | - | - | - |
| GPTQ | W4A8 | W4A4 | 0 | 12 | – | – | – | – | – |
| MQuant | | | 0 | 82 | 9 | 0 | 7.39 | 6.2 | 12.91 |
| Ours | | | 56.85 | 1352 | 452 | 51.75 | 62.14 | 31.7 | 62.09 |
| RTN | | | - | - | - | - | - | - | - |
| GPTQ | W4A4 | W4A4 | – | – | – | – | – | – | – |
| MQuant | | | 0 | 28 | 1 | 0.04 | 5.05 | 5.2 | 11.76 |
| Ours | | | 55.72 | 1322 | 426 | 50.04 | 61.52 | 32.2 | 61.94 |

## 4.1 OVERALL RESULTS

Table 1 reports the overall performance of different quantization methods across seven multimodal benchmarks. Our method consistently outperforms RTN and GPTQ under all bit-width configurations, and remains competitive with or superior to MQuant. Notably, even under low-bit settings such as W4A8 and W4A4, our approach achieves results close to the BF16 baseline, demonstrating strong robustness and generalization ability in aggressive quantization configs.

## 4.2 ABLATION STUDY

We evaluate the proposed forward-based modality weighting on TextVQA using Qwen-VL-Chat and InternVL2 under the vision W8A8 + language W4A8 setting. With all other settings fixed, we compare three conditions: (i) text-only c4 calibration, (ii) COCO without weight, and (iii) COCO with weight. The results show that the modality-weighted strategy consistently achieves higher accuracy than both text calibration and the unweighted COCO setting. This indicates that when formulating the optimization objective, it is crucial to prioritize the fidelity of textual tokens while still accounting for the contribution of visual tokens.

Table 2: Ablation study on modality weight

| Model | Calib | mod.weight | TextVQA |
|-------|-------|------------|---------|
| Qwen-VL | bf16 | – | 60.62 |
| | c4 | – | 59.65 |
| | COCO | no | 59.57 |
| | COCO | yes | 60.18 |
| InternVL2 | bf16 | – | 77.69 |
| | c4 | – | 77.23 |
| | COCO | no | 77.11 |
| | COCO | yes | 77.58 |

## 5 CONCLUSION

Our work presents a computation-invariant PTQ framework for vision–language models that extends rotation-based calibration across the entire stack, including the vision encoder, the projector, and the language decoder. By adopting RMSNorm to preserve orthogonal and scaling invariances, the proposed design enables learned reparameterizations to be fully folded into model parameters at inference, thereby retaining the original forward graph and incurring no runtime overhead. On top of this foundation, we introduce a forward-based, modality-aware weighting scheme for the language branch that reconciles the scale disparity and sensitivity gap between visual and textual tokens during calibration.

Empirically, the quantized models consistently outperform RTN and GPTQ and remain competitive with or superior to MQuant under common bit-width configurations (W8A8, W4A8, W4A4), approaching BF16 accuracy even in low-bit regimes. Ablation on TextVQA under the vision W8A8 + language W4A8 setting shows that enabling modality weighting yields higher accuracy than both text-only calibration and unweighted multimodal calibration, confirming that prioritizing textual fidelity while accounting for visual tokens is an effective objective for VLM PTQ.

## 6 LIMITATIONS AND FUTURE WORK

Although the proposed computation-invariant, modality-aware PTQ framework demonstrates stable gains across benchmarks, two primary limitations remain. First, our experiments are limited to Qwen-VL-Chat and InternVL2-8B, which do not capture the challenges posed by larger-parameter VLMs. In future work, we plan to extend evaluations to larger-scale models and a broader range of families (including larger variants of the same series and other widely adopted open-source VLMs) to validate the generality and robustness of our approach. Second, our method leverages FlatQuant's rotation–reconstruction mechanism to mitigate modality-specific quantization errors; however, introducing learnable parameters and reconstruction optimization inevitably incurs additional computation and memory traffic, which may increase inference latency. Looking forward, we aim to pursue system-level optimizations, including reducing calibration and folding overhead, co-designing with hardware and kernels (e.g., fewer rotation parameters, layout-aware operator fusion), and jointly quantizing activations and KV caches while preserving accuracy, to achieve a more balanced trade-off among accuracy, latency, and energy efficiency for practical deployment.

# 7 ETHICS STATEMENT

This work adheres to the ICLR Code of Ethics. We conducted no studies involving human subjects or animals and did not collect, use, or release any personally identifiable information (PII). Images that may include people were used solely for non-commercial academic research; we did not add or modify annotations or attempt to identify individuals. We evaluate Qwen-VL-Chat and InternVL2-8B under their licenses; our method is post-training quantization (PTQ) without additional fine-tuning or behavior modification, and we did not disable or circumvent any safety mechanisms. All experiments were performed offline and no interactive systems were deployed.

We have taken care to avoid any biases or discriminatory outcomes in our research process. No personally identifiable information was used, and no experiments were conducted that could raise privacy or security concerns. We are committed to maintaining transparency and integrity throughout the research process.

# 8 REPRODUCIBILITY STATEMENT

We have made every effort to ensure that the results presented in this paper are reproducible. All code have been made publicly available in an anonymous repository to facilitate replication and verification. To reproduce the experiments, simply execute the scripts included in the project. We have also provided a full description of our contributions to assist others in reproducing our experiments.

Additionally, the public datasets used in this paper, such as COCO , c4 , and the evaluation suites TextVQA, MME, OCRBench, DocVQA, ScienceQA, MMMU, and SEEDBench, are all publicly available, ensuring consistent and reproducible evaluation results.

We believe these measures will enable other researchers to reproduce our work and further advance the field.

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

# A APPENDIX

## A.1 LLM USAGE

Large Language Models (LLMs) were used to aid in the writing and polishing of the manuscript. Specifically, we used an LLM to assist in refining the language, improving readability, and ensuring clarity in various sections of the paper. The model helped with tasks such as sentence rephrasing, grammar checking, and enhancing the overall flow of the text.

It is important to note that the LLM was not involved in the ideation, research methodology, or experimental design. All research concepts, ideas, and analyses were developed and conducted by the authors. The contributions of the LLM were solely focused on improving the linguistic quality of the paper, with no involvement in the scientific content or data analysis.

The authors take full responsibility for the content of the manuscript, including any text generated or polished by the LLM. We have ensured that the LLM-generated text adheres to ethical guidelines and does not contribute to plagiarism or scientific misconduct.

A.2  PROOF OF THEOREM 1

**Theorem 1 (Principle of Objective De-emphasis)** *Let $L_A(\Theta)$ and $L_B(\Theta)$ be two non-negative loss functions parameterized by $\Theta \in \mathbb{R}^d$. Consider the composite objective function $J(\Theta, \alpha) = L_A(\Theta) + \alpha L_B(\Theta)$, where $\alpha \geq 0$ is a scalar weight.*

*Let $\alpha_1$ and $\alpha_2$ be two weights such that $\alpha_1 > \alpha_2 \geq 0$. Let $\Theta_1 = \arg\min_\Theta J(\Theta, \alpha_1)$ and $\Theta_2 = \arg\min_\Theta J(\Theta, \alpha_2)$, assuming such minimizers exist.*

*Then the following trade-off relationship holds:*

$$L_A(\Theta_2) \leq L_A(\Theta_1) \tag{16}$$
$$L_B(\Theta_1) \leq L_B(\Theta_2) \tag{17}$$

**Proof 1** *The proof is founded on the fundamental definition of the $\arg\min$ operator, which states that a minimizer yields a value for its objective function that is less than or equal to the value produced by any other parameter set.*

*By the definition of $\Theta_1$ as the minimizer of $J(\Theta, \alpha_1)$, its objective value must be less than or equal to that obtained using any other parameter set, including $\Theta_2$. This gives us our first foundational inequality:*

$$J(\Theta_1, \alpha_1) \leq J(\Theta_2, \alpha_1) \implies L_A(\Theta_1) + \alpha_1 L_B(\Theta_1) \leq L_A(\Theta_2) + \alpha_1 L_B(\Theta_2). \tag{18}$$

*Similarly, by the definition of $\Theta_2$ as the minimizer of $J(\Theta, \alpha_2)$, its objective value must be less than or equal to that obtained using any other parameter set, including $\Theta_1$. This provides our second foundational inequality:*

$$J(\Theta_2, \alpha_2) \leq J(\Theta_1, \alpha_2) \implies L_A(\Theta_2) + \alpha_2 L_B(\Theta_2) \leq L_A(\Theta_1) + \alpha_2 L_B(\Theta_1). \tag{19}$$

*These two inequalities encapsulate the entire behavior of the optimization process. We can uncover the trade-off by combining them. Let us add inequality equation 18 and inequality equation 19:*

$$\big(L_A(\Theta_1) + \alpha_1 L_B(\Theta_1)\big) + \big(L_A(\Theta_2) + \alpha_2 L_B(\Theta_2)\big) \leq \big(L_A(\Theta_2) + \alpha_1 L_B(\Theta_2)\big) + \big(L_A(\Theta_1) + \alpha_2 L_B(\Theta_1)\big).$$

*The terms $L_A(\Theta_1)$ and $L_A(\Theta_2)$ appear on both sides and can be cancelled, simplifying the expression to:*

$$\alpha_1 L_B(\Theta_1) + \alpha_2 L_B(\Theta_2) \leq \alpha_1 L_B(\Theta_2) + \alpha_2 L_B(\Theta_1).$$

*Now, we gather the terms involving $L_B(\Theta_1)$ on the left side and terms involving $L_B(\Theta_2)$ on the right side:*

$$\alpha_1 L_B(\Theta_1) - \alpha_2 L_B(\Theta_1) \leq \alpha_1 L_B(\Theta_2) - \alpha_2 L_B(\Theta_2).$$

*Factoring out the common terms yields:*

$$(\alpha_1 - \alpha_2) L_B(\Theta_1) \leq (\alpha_1 - \alpha_2) L_B(\Theta_2).$$

*According to the theorem's premise, we have $\alpha_1 > \alpha_2$, which means the term $(\alpha_1 - \alpha_2)$ is strictly positive. We can therefore divide both sides of the inequality by this positive scalar without altering the direction of the inequality:*

$$L_B(\Theta_1) \leq L_B(\Theta_2).$$

*This proves the second part of the theorem's conclusion, as stated in Eq. equation 17. This intermediate result is highly intuitive: placing a stronger penalty on $L_B$ (by using a larger weight $\alpha_1$) results in a solution $\Theta_1$ that yields a lower or equal error on $L_B$ compared to the solution $\Theta_2$ optimized with a weaker penalty.*

*We now use the trade-off relationship we just derived to prove the main thesis. Let us revisit the second foundational inequality equation 19:*

$$L_A(\Theta_2) + \alpha_2 L_B(\Theta_2) \leq L_A(\Theta_1) + \alpha_2 L_B(\Theta_1).$$

*Rearranging this to isolate the difference between the $L_A$ terms, we get:*

$$L_A(\Theta_2) - L_A(\Theta_1) \leq \alpha_2 L_B(\Theta_1) - \alpha_2 L_B(\Theta_2).$$

*Factoring out $\alpha_2$ on the right-hand side gives:*

$$L_A(\Theta_2) - L_A(\Theta_1) \leq \alpha_2 \big(L_B(\Theta_1) - L_B(\Theta_2)\big).$$

*Let us analyze the sign of the right-hand side.*

1. *From the theorem's premise, we have $\alpha_2 \geq 0$.*

2. *From the result of Step 2, we know $L_B(\Theta_1) \leq L_B(\Theta_2)$, which implies that the term $\big(L_B(\Theta_1) - L_B(\Theta_2)\big)$ is non-positive (i.e., $\leq 0$).*

*The product of a non-negative number ($\alpha_2$) and a non-positive number ($\big(L_B(\Theta_1) - L_B(\Theta_2)\big)$) must itself be non-positive. Therefore:*

$$\alpha_2\big(L_B(\Theta_1) - L_B(\Theta_2)\big) \leq 0.$$

*This directly implies that the left-hand side must also be non-positive:*

$$L_A(\Theta_2) - L_A(\Theta_1) \leq 0.$$

*This leads to the final result of the theorem as stated in Eq. equation 16:*

$$L_A(\Theta_2) \leq L_A(\Theta_1).$$

*This completes the proof, formally demonstrating that reducing the weight on a secondary objective ($L_B$) allows the optimizer to find a solution that is provably no worse—and potentially better—for the primary objective ($L_A$).*

$\square$

