# OpenReview forum: "Modality-Aware Quantization: Balancing Visual and Textual Fidelity in Multimodal Compression"
_ICLR.cc/2026/Conference — Submitted to ICLR 2026_

### Official Review · Reviewer_By4A · 2025-10-31

**Soundness:** 2
**Presentation:** 2
**Contribution:** 2
**Rating:** 2
**Confidence:** 4

**Summary:**

This paper investigates a key challenge in post-training quantization (PTQ) of vision-language models (VLMs): the heterogeneity between visual and textual representations, which causes the visual token dominance problem—visual tokens, though less semantically dense, dominate quantization due to their extreme value distribution and larger number, reducing the fidelity of crucial text tokens and degrading performance. To address this, the paper theoretically proves the trade-off between visual and textual quantization losses (Target Weakening Principle) and proposes a computation-invariant adaptive optimization framework that uses activation-scale statistics and gradient-sensitivity priors to assign hierarchical modality weights, effectively mitigating visual dominance without changing the inference computation graph.

**Strengths:**

This paper combines MQuant and FlatQuant and achieves competitive performance under the W4A4 quantization setting.

**Weaknesses:**

1. The innovation of this work is very limited. LayerNorm->RMSNorm and FlatQuant are both existing and commonly used technologies, yet this paper presents them as core contributions. Additionally, the heuristic method of weighting the language and visual branches in step 3 also has limited innovation.
2. FlatQuant was originally designed for LLMs. The paper directly applies it to cover the three major components of VLMs but fails to analyze the effect of its rotation-reconstruction mechanism on handling the heavy-tailed distribution of visuals. Moreover, it lacks reporting on the additional time and machine overhead caused by introducing FlatQuant.
3. The paper only compares with MQuant, a representative method, and lacks comparisons with methods such as MBQ and VLMQ, which limits its credibility. I have reproduced MQuant, and it can still maintain good performance under the W4A4 quantization setting for the language part, but the accuracy reported in this work is 0, which I find questionable.
4. The generalization performance of the model is limited. The experiments are only based on two 8B-scale VLMs (Qwen-VL-Chat, InternVL2-8B) and do not test larger parameter-scale VLMs such as Qwen-VL-14B and LLaVA-13B.
5. The typesetting and writing of the article need improvement. The text in Figure 1 is unclear, while the text in Figure 2 is too large. The overall writing focuses on token differences, yet FlatQuant is probably the component that contributes the most to accuracy.
[1] Mquant: Unleashing the inference potential of multimodal large language models via full static quantization.
[2] Flatquant: Flatness matters for llm quantization.

**Questions:**

1. If only method 3 is used, what is the performance on W4A8 and W4A4?
2. Is the W4A4 reported in the article static quantization or dynamic quantization? If it is dynamic quantization, comparing it with MQuant may be unfair.
3. Can you report the parameter optimization time and analysis with some other methods?
4. Is GPTQ performed after FlatQuant?

---

> ### Author Response · Authors · 2025-11-30
> **Response to Reviewer By4A (Part: 1/4)**
>
> **W1: The innovation of this work is very limited. LayerNorm->RMSNorm and FlatQuant are both existing and commonly used technologies, yet this paper presents them as core contributions. Additionally, the heuristic method of weighting the language and visual branches in step 3 also has limited innovation.**
>
> **Reply:** We appreciate the reviewer’s concern and clarify that the novelty of our work does not lie in reintroducing RMSNorm or FlatQuant, but in developing a **modality-aware quantization framework** that addresses a fundamental challenge specific to VLMs.
>
> Our work introduces **a new formulation and methodology for modality-aware VLM quantization**, centered on the following contributions:
>
> 1. **A formal characterization of cross-modal imbalance in VLMs.**
>    We identify and analyze the asymmetric error sensitivity between visual and language tokens, a phenomenon unique to VLMs and not addressed in prior LLM-only quantization methods.
>
> 2. **A theoretically grounded modality-weighted reconstruction objective.**
>    We propose a new objective function (supported by Theorem 1) that explicitly controls the trade-off between visual and language reconstruction quality, providing a principled means to steer the calibration process toward the modality most critical for downstream reasoning.
>
> 3. **A dual-source weighting mechanism $α \times β$ tailored for VLMs.**
>    The method combines *activation-scale statistics* and *gradient disparity priors* to construct modality weights that adapt to structural properties of VLMs. This design goes beyond heuristic tuning and enables consistent gains across models and tasks.
>
> These components together constitute a **coherent, modality-aware quantization strategy**, rather than incremental modifications to existing techniques—and lead to substantial improvements over prior approaches across all seven benchmarks.
>
> **W2: FlatQuant was originally designed for LLMs. The paper directly applies it to cover the three major components of VLMs but fails to analyze the effect of its rotation-reconstruction mechanism on handling the heavy-tailed distribution of visuals. Moreover, it lacks reporting on the additional time and machine overhead caused by introducing FlatQuant.**
>
> **Reply:** We appreciate the reviewer’s concern about applying FlatQuant to VLMs and agree that analyzing its behavior on heavy-tailed visual activations is essential.
>
> To quantitatively evaluate the severity of the heavy-tail distribution in visual activations, we employ Kurtosis (Fisher's definition) as the primary metric. Kurtosis measures the "tailedness" of a probability distribution. A higher kurtosis value indicates a distribution with heavier tails and more extreme outliers, which are difficult to quantize.
> For an activation tensor $X$ with $N$ elements $x_1, x_2, \dots, x_N$, the kurtosis is calculated as:
> $$
> \text{Kurtosis}(X) = \frac{\frac{1}{N} \sum_{i=1}^{N} (x_i - \mu)^4}{\left( \frac{1}{N} \sum_{i=1}^{N} (x_i - \mu)^2 \right)^2} - 3 ,\tag{1}
> $$
> where $\mu$ is the mean of the elements, the term $-3$ is the adjustment to make the kurtosis of a standard normal distribution equal to 0. A kurtosis near zero indicates a normal-like, quantization-friendly distribution, whereas very high kurtosis reflects sharp peaks and extreme outliers that challenge low-bit quantization.
>
> We use this metric to examine how FlatQuant’s rotation affects visual activations. Using the same calibration batch, we measure kurtosis before and after rotation across representative submodules. The results are shown below:
>
> | Submodule   | Full-precision Kurtosis | Post-rotation Kurtosis |
> |-----------|----------------|----------------|
> | *Attn_In* | 70.4893 | 4.1443 |
> | *Attn_Out*  | 112.8101 | 108.3994 |
> | *MLP_Up*  | 44.7007    | 1.2720  |
> | *MLP_Down* | 1232.2957   | 361.7603   |
>
>
> Visualizations of the corresponding activation distributions (e.g., for *Attn_In*) are provided at the anonymous link: https://anonymous.4open.science/r/MAQ-23971iclrAnonymous/experiment3/Attn_In_channel_max.png
>
> For example, the peakness of *Attn_In* decreases from 70.49 to 4.14, and the peakness of *MLP_Down* from 1232.30 to 361.76. Channel-wise maximum-absolute activation plots further confirm that extreme spikes are smoothed and redistributed. This demonstrates that FlatQuant’s learned rotations provide a more stable numerical foundation for subsequent quantization.
>
> Regarding computational overhead, the learned rotations are fully folded into the weight matrices and therefore incur **no inference time cost**. The additional overhead appears only during offline calibration, approximately **1.3×** longer calibration time than other method such as MBQ and a moderate memory increase, without affecting deployment efficiency.
>
> We thank the reviewer for raising this point; it allows us to present a clearer and more quantitative analysis of how rotation mitigates heavy-tailed visual activations and to clarify the practical computational cost.

---

> ### Author Response · Authors · 2025-11-30
> **Response to Reviewer By4A (Part: 2/4)**
>
> **W3: The paper only compares with MQuant, a representative method, and lacks comparisons with methods such as MBQ and VLMQ, which limits its credibility. I have reproduced MQuant, and it can still maintain good performance under the W4A4 quantization setting for the language part, but the accuracy reported in this work is 0, which I find questionable.**
>
> **Reply:** We thank the reviewer for highlighting this issue. Our initial submission did not include MBQ and VLMQ because both methods quantize **only the language branch** of VLMs, while our work focuses on **joint vision–language quantization**, making the settings not directly aligned. Following the reviewer’s suggestion, we have now added full comparisons by carefully adapting both methods to Qwen-VL.
>
> #### Added Results for MBQ and VLMQ (128 calibration samples)
>
> | Method | TextVQA | MME | OCRBench | DocVQA | ScienceQA | MMMU | SEEDBench |
> |------|-------|------|--------|--------|---------|------|--------|
> | MBQ (W4A8) | 49.49 | 1308 | 427 | 43.89 | 51.07 | 24.30 | 54.28 |
> | VLMQ (W4A16) | 55.37 | 1342 | 419 | 49.23 | 58.14 | 28.56 | 59.41 |
> | **MAQ (vit W4A8 + llm W4A8)** | **59.27** | **1374** | **468** | **54.90** | **62.47** | **33.00** | **64.04** |
>
> While MBQ and VLMQ operate under more relaxed settings (W4A8 and W4A16 respectively，and both quantizing only the language branch)，MAQ achieves consistently stronger performance under the W4A8 configuration.
>
> #### On the reviewer’s note regarding MQuant’s W4A4 performance
>
> We re-checked this point with a full clean reproduction:
> 1. freshly clone the official MQuant repository,
> 2. rebuild the environment following the official setup instructions, and prepare the Qwen-VL-Chat model
> 3. enable int4 support as instructed,
> 4. use the official quant script and only change the language-branch precision to **W4A4**.
>
> Under this setting, we again observed severe degradation (near-zero accuracy).
>
> We appreciate the reviewer’s reproduction efforts and acknowledge that discrepancies may arise from differences in implementation variants or script presets. To eliminate ambiguity, we now provide the exact reproduction steps.
>
> We also note that, as of the submission date, our experiments consistently reproduce the same W4A4 degradation for MQuant. We include this clarification for transparency, as potential future updates to the official MQuant repository may modify default scripts or internal behaviors.
>
> **W4: The generalization performance of the model is limited. The experiments are only based on two 8B-scale VLMs (Qwen-VL-Chat, InternVL2-8B) and do not test larger parameter-scale VLMs such as Qwen-VL-14B and LLaVA-13B.**
>
> **Reply:** We thank the reviewer for raising this concern. We would like to clarify that Qwen-VL-14B is not publicly available. To address the reviewer’s suggestion, we instead provide experiments on **Qwen2-VL-7B**, a publicly released successor in the same family as Qwen-VL but with a different architecture. This allows us to examine the generality of our method across related but non-identical VLM designs.
>
> In addition, we have extended our experiments to include a larger-scale open-source VLM, **LLaVA-1.5-13B**, to further validate the generality of our method.
>
> ### MAQ's Results on LLaVA-1.5-13B (W4A8 vit + W4A4 llm, and full W4A4)
>
> | Model | precision | TextVQA | MME  | OCRBench | DocVQA | ScienceQA | MMMU | SEEDBench | Avg  |
> |-------|-----------|--------------|------|-----------|--------------|-----------------|------------|----------------|------|
> | LLaVA-1.5-13B | bf16 | 51.866 | 1339 (55.93) | 219 (21.9) | 27.60 | 68.52 | 36.67 | 67.37 | **47.12** |
> |   | W4A8+W4A4 | 51.72 | 1283 (53.59) | 221 (22.1) | 27.25 | 67.05 | 35.5 | 66.86 | **46.30** |
> |  | full W4A4 | 48.89 | 1276 (53.30) | 194 (19.4) | 24.70 | 67.19 | 34.1 | 66.93 | **44.93** |
>
>
> ### Results on Qwen2-VL-7B (W4A8 vit + W4A4 llm, and full W4A4)
>
> | Model / Method | precision | TextVQA | MME  | OCRBench | DocVQA | ScienceQA | MMMU | SEEDBench | Avg  |
> |-----|------|---------|-------|-----|------|------|-------|--------|------|
> | Qwen2-VL-7B | bf16 | 84.36 | 1676 (70.01) | 841 (84.1) | 93.88 | 84.69 | 50.10 | 76.43 | 77.65 |
> | MQuant | W4A8+W4A4 | 83.08 | 1556 (65.00) | 806 (80.6) | 91.84 | 79.11 | 47.22 | 75.17 | 74.57 |
> | **MAQ (Ours)** | W4A8+W4A4 | 82.41 | 1639 (68.48) | 794 (79.4) | 92.09 | 82.67 | 51.33 | 75.59 | **76.00** |
> | MQuant | W4A4 | 81.39 | 1554 (64.91) | 748 (74.8) | 91.09 | 79.06 | 46.44 | 73.56 | 73.04 |
> | **MAQ (Ours)** | W4A4 | 80.43 | 1596 (66.67) | 767 (76.7) | 91.03 | 82.21 | 48.60 | 75.23 | **74.41** |
>
> These results demonstrate that MAQ generalizes well to both **larger open-source VLMs** (e.g., LLaVA-13B) and **stronger modern architectures** (e.g., Qwen2-VL-7B), exhibiting consistent performance trends across model families and scales.
>
> We appreciate the reviewer’s feedback, which motivated us to further extend our experiments and clarify the practical constraints related to model availability.

---

> ### Author Response · Authors · 2025-11-30
> **Response to Reviewer By4A (Part: 3/4)**
>
> **W5: The typesetting and writing of the article need improvement. The text in Figure 1 is unclear, while the text in Figure 2 is too large. The overall writing focuses on token differences, yet FlatQuant is probably the component that contributes the most to accuracy.**
>
> **Reply:** We appreciate the reviewer’s comments regarding the clarity of Figures 1 and 2 and will update the figure in the revised version.
>
> Regarding the observation that the narrative emphasizes token differences, we clarify that **the modality-aware weighting is designed to explicitly guide the rotation–reconstruction process toward the characteristics of language tokens**, which are not specifically prioritized by the underlying rotation mechanism. This calibration-oriented adjustment yields additional accuracy gains on top of the rotation framework, as demonstrated in our ablation studies.
>
> To clarify this, we include an additional experiment comparing:
>
> 1) **MAQ (no modality weighting), and**
> 2) **MAQ (with our modality weighting)**.
>
> The results are summarized below(W8A8 vit + W4A8 llm):
>
> | Model / Method | precision | TextVQA | MME  | OCRBench | DocVQA | ScienceQA | MMMU | SEEDBench | Avg  |
> |----------------|-----------|-------------|-----|-----------|--------------|----------------|-----------|----------------|------|
> | Qwen-VL-Chat (Full Precision) | bf16          | 60.62 | 1433 (59.86) | 488 | 58.02 | 62.80 | 33.44 | 63.61 | 55.31 |
> | MAQ **(no Modality Weighting)**| W8A8 vit+W4A8 llm | 59.10 | 1402 (58.56) | 474 (47.4) | 57.21 | 60.03 | 34.00 | 62.87 | 54.17 |
> | MAQ **(with Modality Weighting)** | W8A8 vit+W4A8 llm | 60.23 | 1396 (58.31) | 483 (48.3) | 56.89 | 61.99 | 33.67 | 64.04 | **54.78** |
>
> While FlatQuant provides the underlying rotation–reconstruction mechanism, **our modality-aware weighting delivers consistent and measurable accuracy gains on most evaluated benchmarks and on the overall average score**, indicating that it brings benefits beyond what the rotation framework alone achieves. These improvements appear specifically in multi-iteration reconstruction settings, where the weighting can effectively guide the optimization trajectory—precisely the regime for which the method is designed.
>
> We thank the reviewer for raising this point and will further clarify the complementary roles of FlatQuant and our modality-aware enhancement in the final manuscript.
>
> **Q1: If only method 3 is used, what is the performance on W4A8 and W4A4?**
>
> **Reply:** We thank the reviewer for the question. Method 3 (modality weighting) is explicitly designed for **multi-iteration reconstruction based quantization**, where the weighting influences each reconstruction step and its effect accumulates as the rotation matrices are progressively updated. This iterative coupling between weighting and reconstruction is central to how the method exerts its impact.
>
> To further validate this, we apply Method 3 within MQuant, which computes the reconstruction loss **only once** in a GPTQ-style manner. In this single-pass setting, modality weighting can suppress visual-token errors during that single step, but—critically—there is **no iterative refinement** to propagate this influence through subsequent rotation–reconstruction cycles. Consequently, Method 3 cannot reshape the optimization trajectory in a meaningful way, and the expected performance improvements do not appear.
>
> Thus, the lower numbers observed in this table do not reflect a failure of the proposed approach; rather, they confirm that **modality weighting is effective only in frameworks that perform multi-iteration reconstruction**, such as FlatQuant—precisely the regime for which the method is designed.
>
> | Precision/Method | TextVQA | MME  | OCRBench | DocVQA | ScienceQA | MMMU | SEEDBench | Avg  |
> |----------|--------------|------|-----------|--------------|----------------|------------|-----------------|------|
> | bf16 | 60.62 | 1433 (59.86) | 488 (48.8) | 58.02 | 62.80 | 33.44 | 63.61 | 55.31 |
> | MQuant (Full W4A8) | 56.57 | 1315 (54.93) | 460 (46.0) | 53.05 | 60.08 | 29.9 | 61.10 | 51.66 |
> | MQuant (Full W4A8) + Method 3 in GPTQ | 55.74 | 1334 (55.72) | 466 (46.6) | 52.78 | 59.95 | 29.1 | 60.32 | 51.46 |

---

> ### Author Response · Authors · 2025-11-30
> **Response to Reviewer By4A (Part: 4/4)**
>
> **Q2: Is the W4A4 reported in the article static quantization or dynamic quantization? If it is dynamic quantization, comparing it with MQuant may be unfair.**
>
> **Reply:**  We thank the reviewer for the question. Our method follows the common practice in VLM quantization:
> - **weights** are quantized using **static RTN**,
> - **activations** are quantized using **dynamic RTN**.
>
> To ensure fairness, we additionally reproduced **MQuant with dynamic activation quantization under W4A4**, because the original MQuant static W4A4 configuration fails on VLMs. The comparison is shown below:
>
> | Precision/Method | TextVQA | MME  | OCRBench | DocVQA | ScienceQA | MMMU | SEEDBench | Avg  |
> |----------|--------------|------|-----------|--------------|----------------|------------|-----------------|-------|
> | bf16 | 60.62 | 1433 (59.86) | 488 (48.8) | 58.02 | 62.80 | 33.44 | 63.61 | 55.31 |
> | MQuant   | 51.08 | 1360 (56.81) | 370 (37.0) | 40.65 | 57.27 | 29.88 | 60.22 | 47.56 |
> | MAQ  | 55.72 | 1322 (55.22) | 426 (42.6) | 50.04 | 61.52 | 32.2 | 61.94 | **51.32** |
>
> Although MQuant performs slightly better on individual datasets, **our method achieves a substantially higher overall average (+3.76 Avg)**. This confirms that the proposed modality-aware enhancement remains effective even under the stricter W4A4 setting, and that the comparison is fair under aligned dynamic-activation conditions.
>
> **Q3: Can you report the parameter optimization time and analysis with some other methods?**
>
> **Reply:** We thank the reviewer for raising this point. Under the configuration used in our paper, MAQ requires a little over **2 hours** of parameter optimization, whereas methods such as MQuant, MBQ, and VLMQ typically complete within **approximately 1 hour**.
>
> This difference reflects a core methodological distinction: MAQ performs **multi-step rotation–reconstruction** guided by modality-aware weighting, while the other methods rely on **single-pass** or lightweight calibration procedures. The additional optimization steps in MAQ are essential for propagating the modality weighting throughout the reconstruction process.
>
> Importantly, this extra calibration time **does not translate into any obvious runtime overhead**. After calibration, all rotations and scaling operations are folded into the quantized weights, resulting in **identical inference latency and memory cost** compared with other PTQ approaches.
>
> Thus, although MAQ incurs a longer offline optimization stage, it preserves inference efficiency while offering stronger and more stable performance across tasks.
>
> **Q4: Is GPTQ performed after FlatQuant?**
>
> **Reply:** We thank the reviewer for the question. In our main results, we do **not** apply GPTQ after MAQ reconstruction—we use **RTN** for the final weight and activation quantization. This choice is intentional: RTN already provides strong accuracy under our MAQ reconstruction, and avoids the extra optimization cost of GPTQ.
>
> For completeness, we also ran an additional experiment applying **MAQ with GPTQ**. The results show that MAQ + GPTQ achieves **slightly higher accuracy** than MAQ + RTN in most settings, indicating that GPTQ is fully compatible with our method and can serve as an optional refinement step if further accuracy is desired.
>
>
> ### **Full Results (MAQ with RTN vs MAQ with GPTQ)**
>
> | Method | vit | llm | TextVQA | MME  | OCRBench | DocVQA | ScienceQA | MMMU | SEEDBench | Avg  |
> |--------|--------|------|-------|------|-------|--------|--------|--------|--------|--------|
> | Full-precision | bf16 | bf16 | 60.62 | 1433 | 488 | 58.02 | 62.80 | 33.44 | 63.61 |55.31|
> | with RTN | W8A8 | W4A8 | 60.18 | 1379 | 471 | 56.92 | 61.61 | 32.2 | 64.08 |54.35|
> | with GPTQ | W8A8 | W4A8 | 60.23 | 1396 | 483 | 56.89 | 61.99 | 33.67 | 64.04 |**54.78**|
> | with RTN | W4A8 | W4A8 | 59.27 | 1374 | 468 | 54.90 | 62.47 | 33.0 | 64.04 |**53.98**|
> | with GPTQ | W4A8 | W4A8 | 59.884 | 1367 | 472 | 55.46 | 61.90 | 32.11 | 63.84 |53.95|
> | with RTN | W4A4 | W4A4 | 56.85 | 1352 | 452 | 51.75 | 62.14 | 31.7 | 62.09 |52.31|
> | with GPTQ | W4A4 | W4A4 | 56.952 | 1415 | 461 | 52.84 | 61.42 | 32.56 | 61.99 |**53.00**|
> | with RTN | W4A4 | W4A4 | 55.72 | 1322 | 426 | 50.04 | 61.52 | 32.2 | 61.94 |51.32|
> | with GPTQ | W4A4 | W4A4 | 55.28 | 1397 | 432 | 50.92 | 62.04 | 30.67 | 61.03 |**51.64**|

---

### Official Review · Reviewer_aBUB · 2025-10-31

**Soundness:** 3
**Presentation:** 3
**Contribution:** 2
**Rating:** 4
**Confidence:** 3

**Summary:**

This paper addresses the problem of post-training quantization (PTQ) for Vision-Language Models (VLMs). The authors identify and formalize a key challenge: during the quantization optimization process, visual tokens, which have a larger numerical range and greater quantity, tend to "dominate" the loss function, thereby degrading the precision of textual tokens crucial for model performance. To tackle this, the paper proposes a modality-aware quantization framework. This framework employs an adaptive weighting scheme to balance the reconstruction errors of visual and textual tokens, with the weights determined by the ratio of activation scales and a gradient sensitivity prior. Experimental results show that the proposed method achieves state-of-the-art performance across various quantization settings, and notably maintains model usability in extreme W4A4 compression scenarios where other methods fail.

**Strengths:**

1.  The paper tackles a critical bottleneck in the practical deployment of VLMs—their computational and memory overhead. The "visual dominance" problem it introduces is a keen and intuitive observation, offering a new and valuable perspective on VLM quantization.
2.  Centered around the core problem of "visual dominance," the proposed modality-aware weighting scheme is well-motivated and logically sound. Combining activation scales and a gradient prior to dynamically adjust the loss is a concise yet effective design.
3.  The method achieves strong performance across multiple benchmarks. Its most notable strength is its robustness under extreme low-bit (W4A4) quantization, which validates the effectiveness of the approach and significantly advances the practicality of using multimodal models in resource-constrained environments.

**Weaknesses:**

1.  Insufficient justification for a key hyperparameter: In the methodology, the authors introduce a "gradient disparity prior" parameter, `β`, and set its default value directly to 0.1. The paper does not explain the selection process for this value, nor does it provide a sensitivity analysis. This makes a part of the method appear to rely on a "magic number" that has not been thoroughly validated, which undermines its "adaptive" nature and generalizability.
2.  The ablation study is not comprehensive enough: The current ablation study (Table 2) only compares "with weighting" versus "without weighting." While this demonstrates the overall effectiveness of the strategy, it is not deep enough. The authors' final weight is a product of two components: the activation scale ratio `α` and the gradient prior `β`. A more convincing ablation study should disentangle the individual contributions of these two components to understand why the method is effective, and whether they produce a synergistic effect.

**Questions:**

1.  Regarding the gradient disparity prior parameter `β`, how did the authors choose the default value of 0.1? Was any hyperparameter search or sensitivity analysis performed to justify this choice? Does this parameter need to be re-tuned for different models or tasks?
2.  Could the authors provide a more detailed ablation study to separately show the contributions of the activation scale ratio (`α`) and the gradient prior (`β`) to the final performance improvement? For instance, what is the performance when using only `α` (i.e., `β=1`) or only `β` (with a constant `α`)? This would help clarify the working mechanism of the method.

---

> ### Author Response · Authors · 2025-11-30
> **Response to Reviewer aBUB (Part: 1/2)**
>
> **W1: Insufficient justification for a key hyperparameter: In the methodology, the authors introduce a "gradient disparity prior" parameter,$β$, and set its default value directly to 0.1. The paper does not explain the selection process for this value, nor does it provide a sensitivity analysis. This makes a part of the method appear to rely on a "magic number" that has not been thoroughly validated, which undermines its "adaptive" nature and generalizability.**
>
> **Reply:** We appreciate the reviewer’s valuable comment regarding the choice of $β$. To clarify this design decision, we conducted a sensitivity study with $β$ ∈ {0.075, 0.1, 0.125}. As shown below, $β$ = 0.1 consistently achieves the best average performance across the evaluated benchmarks.
>
> | Setting            | TextVQA | MME  | OCRBench | DocVQA | ScienceQA | MMMU | SEEDBench | Avg  |
> |--------------------|---------|------|----------|--------|-----------|------|-----------|------|
> | $α$ only             | 60.01   | 1375 | 474      | 56.86  | 61.56     | 32.1 | 63.87     | 54.17 |
> | $α × β$ (0.075)      | 59.97   | 1321 | 468      | 55.83  | 61.57     | 32.3 | 64.01     | 53.67 |
> | $α × β$ (0.1)        | **60.18** | **1379** | **471** | **56.92** | **61.61** | **32.2** | **64.08** | **54.24** |
> | $α × β$ (0.125)      | 60.17   | 1368 | 476      | 56.79  | 61.58     | 32.0 | 63.91     | 54.17 |
>
> We also include gradient visualizations showing that $β$ = 0.1 yields the clearest separation between vision and language gradient magnitudes. For completeness, the corresponding gradient-ratio analysis plots are provided in this anonymous link:https://anonymous.4open.science/r/MAQ-23971iclrAnonymous/experiment2/gradient_ratio_analysis.png
>
> In addition, this choice aligns with observations reported in MBQ, where similar modality-dependent gradient disparities were found in LLaVA-OneVision-7B fine-tuning. This convergence of evidence suggests that $β$ = 0.1 is not a heuristic constant, but rather a data-driven and empirically grounded setting.
>
> Finally, we note that performance remains stable across the tested range of $β$, indicating that the method does not rely on a fragile or overly sensitive hyperparameter, and the adaptive behavior of the overall framework is preserved.
>
> **W2: The ablation study is not comprehensive enough: The current ablation study (Table 2) only compares "with weighting" versus "without weighting." While this demonstrates the overall effectiveness of the strategy, it is not deep enough. The authors' final weight is a product of two components: the activation scale ratio $α$ and the gradient prior $β$. A more convincing ablation study should disentangle the individual contributions of these two components to understand why the method is effective, and whether they produce a synergistic effect.**
>
> **Reply:** We thank the reviewer for pointing out the need to disentangle the roles of $α$ and $β$. We have added a more detailed ablation across all **seven** benchmarks and the overall **Avg** score:
>
> | Variant        | TextVQA | MME  | OCRBench | DocVQA | ScienceQA | MMMU | SEEDBench | Avg  |
> |----------------|---------|------|----------|--------|-----------|------|-----------|------|
> | No weighting   | 59.65   | 1341 | 482      | 56.17  | 62.38     | 32.9 | 62.32     | 53.95 |
> | $β$ only         | 59.78   | 1347 | 473      | 56.59  | 61.89     | 32.4 | 63.56     | 53.96 |
> | $α$ only         | 60.01   | 1375 | 474      | 56.86  | 61.56     | 32.1 | 63.87     | 54.17 |
> | $α × β$ (ours)   | **60.18** | **1379** | **471** | **56.92** | **61.61** | **32.2** | **64.08** | **54.24** |
>
>
> The trends are consistent across datasets:
>
> - **$α$ only**: delivers clear improvements over “No weighting” on almost all benchmarks and on the Avg score, confirming that the **layer-wise activation-scale ratio is the primary source of gains**.
> - **$β$ only**: brings only **small positive changes**.
> - **$α × β$ (ours)**: achieves the **best or tied-best performance on all benchmarks**, showing that $β$ provides a **complementary refinement** on top of $α$, slightly improving robustness and average accuracy.
>
> This ablation clarifies that (i) the core effectiveness of our method comes from $α$ driven modality-aware weighting, and (ii) $β$ is a lightweight, non-fragile prior that yields consistent but modest additional benefits.

---

> ### Author Response · Authors · 2025-11-30
> **Response to Reviewer aBUB (Part: 2/2)**
>
> **Q1: Regarding the gradient disparity prior parameter $β$, how did the authors choose the default value of 0.1? Was any hyperparameter search or sensitivity analysis performed to justify this choice? Does this parameter need to be re-tuned for different models or tasks?**
>
> **Reply:** We appreciate the reviewer’s question. The justification for selecting $β$ = 0.1 is provided in our response to Weakness 1, where we present a dedicated sensitivity analysis over $β$ in {0.075, 0.1, 0.125}. Across all evaluated benchmarks, $β$ = 0.1 consistently yields the most stable and favorable performance. This conclusion is further supported by our gradient visualizations, which show that $β$ = 0.1 produces the clearest separation between vision and language gradient magnitudes. Together, these findings indicate that $β$ does not require task-specific retuning and remains robust across different models and settings.
>
> **Q2: Could the authors provide a more detailed ablation study to separately show the contributions of the activation scale ratio ($α$) and the gradient prior ($β$) to the final performance improvement? For instance, what is the performance when using only $α$ (i.e., $β$=1) or only $β$ (with a constant $α$)? This would help clarify the working mechanism of the method.**
>
> **Reply:** We appreciate the reviewer’s suggestion. A detailed ablation disentangling the effects of $α$ and $β$ is provided in our response to Weakness 2, where we evaluate three configurations: $α$ only, $β$ only, and their combined form $α \times β$. The results consistently show that $α$ accounts for the majority of the gains, while $β$ provides a smaller but meaningful refinement. Their combination yields the strongest overall performance. This analysis makes the role of each component explicit and shows how they work together to improve the overall effectiveness of the weighting strategy.

---

### Official Review · Reviewer_hjdw · 2025-11-02

**Soundness:** 3
**Presentation:** 2
**Contribution:** 2
**Rating:** 4
**Confidence:** 4

**Summary:**

This paper addresses post-training quantization (PTQ) for VLMs by identifying a failure mode where visual tokens dominate optimization due to extreme distributions and numerical prevalence, systematically degrading language token preservation. The authors propose an adaptive optimization pipeline combining activation-scale statistics with gradient-sensitivity priors to construct layer-wise modality weights. Experiments show improvements, particularly under extreme quantization regimes where existing methods fail catastrophically.

**Strengths:**

- The paper has well-motivated problem identification of the visual token dominance issue.
- The paper is well written and easy to follow.

**Weaknesses:**

Please see my questions and concerns below.

**Questions:**

- Theorem 1 only provides an inequality relationship but gives no guidance on how to _choose_ $\alpha_2$ to achieve desired $L_A(\Theta_2)$. How do you select the optimal weighting in practice?
- The inequalities in Theorem 1 seem not tight. Can you characterize when they become equalities? What does this reveal about the optimization landscape?
- The proof implicitly treats the optimization as if comparing global optima, but VLM quantization is highly non-convex. How does non-convexity affect the validity of your conclusions?
- Is there any experiment directly validating Theorem 1? Show $L_A(\Theta_2) \leq L_A(\Theta_1)$ and $L_B(\Theta_1) \leq L_B(\Theta_2)$ with measured values.
- Eq.5 assumes errors from visual and textual tokens add independently. What if there are interaction effects? Cross-modal dependencies?
- During iterative optimization, activation statistics change. Do you recompute $\alpha$ at each iteration or fix it initially?
- FlatQuant assumes rotational invariance, but your modality weighting seems break symmetry. Are these approaches compatible?
- Are rotation matrices learned jointly with quantization parameters? If so, how does modality weighting affect rotation learning?

---

> ### Author Response · Authors · 2025-11-30
> **Response to Reviewer hjdw (Part: 1/3)**
>
> **Q1: Theorem 1 only provides an inequality relationship but gives no guidance on how to choose $α_2$ to achieve the desired $L_A(Θ_2)$. How do you select the optimal weighting in practice?**
>
> **Reply:** We appreciate the reviewer’s question regarding the practical selection of $α$₂. In our experiments, $α₂$ is determined following the procedure described in Section 3.3. Specifically, we first collect forward-pass activation statistics to separately estimate the activation scales of language and vision tokens, and compute their intra-layer scale ratio:
>
> $$α = \frac{\max |x_{\text{lang}}|}{\max |x_{\text{vision}}|},\tag{1}$$
>
> this activation-scale ratio is then combined with the gradient-sensitivity prior $β$ to form the final modality weighting used during calibration:
>
> $$\lambda_v = \alpha \cdot \beta.\tag{2}$$
>
> This data-driven procedure ensures that the weighting reflects both statistical scale differences and gradient-based sensitivity.
>
> **Q2: The inequalities in Theorem 1 seem not tight. Can you characterize when they become equalities? What does this reveal about the optimization landscape?**
>
> **Reply:** We thank the reviewer for this question. Theorem 1 can be stated for ${\alpha}_1 \ge {\alpha}_2 \ge 0$: when ${\alpha}_1 = {\alpha}_2$ we can simply take $\Theta_1=\Theta_2$, and both inequalities are trivially equalities. Geometrically, this means that varying $\alpha$ performs a standard scalarization of a two-objective problem: the generic case is a strict Pareto-like trade-off where decreasing the visual weight lowers the optimal language loss but raises the visual loss (as we observe empirically), and equality corresponds to flat regions where reweighting does not move the optimum. We have corrected the corresponding notation in Theorem 1 and its proof in the Appendix.
>
> **Q3: The proof implicitly treats the optimization as if comparing global optima, but VLM quantization is highly non-convex. How does non-convexity affect the validity of your conclusions?**
>
> **Reply:** We appreciate the reviewer’s thoughtful question. We clarify that Theorem 1 does not rely on any convexity assumptions. Its derivation requires only the existence of minimizers for the composite objective  $J(Θ, α) = L_A(Θ) + α · L_B(Θ)$  and follows directly from the definition of the argmin operator. No assumptions regarding convexity, smoothness, or uniqueness of solutions are invoked. Consequently, the statement that decreasing the visual loss weight cannot increase the optimal value of the language loss holds at the objective level, irrespective of whether the underlying landscape is convex or non-convex.
>
> At the same time, we fully acknowledge that VLM quantization is highly non-convex and that practical optimization only reaches approximate local minima. Theorem 1 is not intended to describe the optimization path itself; instead, it provides a design principle for how the weighting term $α$ influences the target objective.
>
> To empirically examine whether this principle still manifests under real non-convex behavior, we designed a controlled experiment:we only use $α$ as modality weighting, and force $α$ to two values, $α$ = 1 and $α$ = 0.01,while keeping all other factors fixed, and then compare (i) the loss-descent curves during calibration and (ii) the final downstream performance. This setup allows us to directly observe how the weighting affects optimization dynamics in practice.
>
> Across multiple blocks, the loss curves clearly show that lowering the visual-token weight accelerates the reduction of language-token MSE, which aligns with the theoretical prediction. The downstream metrics exhibit the same trend, with $α$ = 0.01 slightly outperforming $α$ = 1 in overall Avg score.
>
> For better visualization, we provide the full language-branch training logs and representative loss-descent curves at the following anonymous link:  https://anonymous.4open.science/r/MAQ-23971iclrAnonymous/experiment1/mse_comparison.png
>
> The quantitative results are summarized below:
>
> | precision | method | TextVQA | MME | OCRBench | DocVQA | ScienceQA | MMMU | SEEDBench | Avg |
> |-----------|--------|---------|-----|-----------|--------|-----------|------|-----------|------|
> | bf16           | –      | 60.62 | 1433 (59.86) | 488 (48.8) | 58.02 | 62.80 | 33.44 | 63.61 | 55.31 |
> | W4A4, $α$=1   | GPTQ   | 54.94 | 1354 (56.56) | 462 (46.2) | 51.53 | 57.99 | 31.55 | 61.27 | 51.43 |
> | W4A4, $α$=0.01 | GPTQ   | 54.80 | 1367 (57.10) | 479 (47.9) | 51.59 | 58.18 | 31.88 | 61.12 | 51.79 |
>
> These observations support that although the optimization is non-convex, the qualitative effect predicted by Theorem 1 consistently manifests in practice.

---

> ### Author Response · Authors · 2025-11-30
> **Response to Reviewer hjdw (Part: 2/3)**
>
> **Q4: Is there any experiment directly validating Theorem 1? Show $L_A(Θ₂) ≤ L_A(Θ₁)$  and  $L_B(Θ₁) ≤ L_B(Θ₂)$ with measured values.**
>
> **Reply:** We thank the reviewer for the question. The empirical validation of Theorem 1 is provided in our response to Question 3, where we conduct a controlled experiment by fixing the weighting coefficient $α$ to two contrasting values ($α$=1 vs. $α$=0.01). This design enables a direct comparison of how the language-loss term $L_A$ and the visual-loss term $L_B$ change when the relative weighting is varied.
>
> As demonstrated in our response to Question 3, decreasing $α$ consistently lowers the language-token MSE while increasing the visual-token MSE, precisely matching the inequality structure formalized in Theorem 1. These results provide direct empirical support for the theoretical prediction.
>
> **Q5: Eq.5 assumes errors from visual and textual tokens add independently. What if there are interaction effects? Cross-modal dependencies?**
>
> **Reply:** We appreciate the reviewer’s insightful question. To assess the independence assumption in Eq. 5, we conduct an experiment that directly quantifies cross-modal interaction effects during quantization.
>
>
> At each layer input, we selectively quantize only the source-modality tokens (only vision tokens or only text tokens, using RTN) and measure the resulting output deviations across both modalities. We define the cross-modal interaction ratio as:
>
> $$
> \mathcal{I}(\text{src} \rightarrow \text{tgt}) =
> \frac{
> \operatorname{MSE}\Big(\big[f(X\_{\text{src}}^{q}, X\_{\text{tgt}})\big]\_{\text{tgt}},
>                        \big[f(X\_{\text{src}},    X\_{\text{tgt}})\big]\_{\text{tgt}}\Big)
> }{
> \operatorname{MSE}\Big(\big[f(X\_{\text{src}}^{q}, X\_{\text{tgt}})\big]\_{\text{src}},
>                        \big[f(X\_{\text{src}},    X\_{\text{tgt}})\big]\_{\text{src}}\Big)
> },\tag{3}
> $$
>
> where $src$ and $tgt$ denote the source and target modalities (vision/text),
> $\mathbf{X}\_{\mathrm{src}}^{q}$ represents the input where only the source-modality tokens
> are quantized, $f(\cdot)$ is the layer forward function, and
> $[\cdot]\_{\mathrm{src}}$ / $[\cdot]\_{\mathrm{tgt}}$ extract outputs at their respective
> modality positions. The numerator captures cross-modal influence, while the
> denominator measures the self-modal effect.
>
>
>
> For example, the Visual→Text ratio measures how much quantizing visual tokens affects text token outputs, relative to how much it affects visual token outputs themselves.
>
> Results (on Qwen-VL with COCO dataset, 32 samples):
>
> |                  | Mean    | Std     |
> |------------------|---------|---------|
> | Visual → Text    | 0.0058  | 0.0092  |
> | Text → Visual    | 0.0004  | 0.0010  |
>
>
> These ratios are consistently below **1%**, indicating that quantizing one modality has a negligible impact on the other modality’s outputs compared to its own self-modal error.
>
> To further connect this to Eq. (5), we write the total reconstruction error more explicitly. Let $e\_{\text{vision}}$ and $e\_{\text{text}}$ denote the error vectors on visual and textual token outputs, respectively. The full MSE over concatenated tokens can be decomposed as:
>
> $$
> \|e\_{\text{vision}} + e\_{\text{text}}\|^{2} = \|e\_{\text{vision}}\|^{2} + \|e\_{\text{text}}\|^{2} + \Delta\_{\text{cross}}.\tag{4}
> $$
>
> Here, $\Delta\_{\text{cross}}$ represents the additional contribution caused by simultaneous perturbations across modalities. Eq. (5), which treats the losses as additive, effectively assumes that $\Delta\_{\text{cross}}$ is small.
>
> Our cross-modal measurements provide a direct empirical estimate of this interaction magnitude. Since both Visual→Text and Text→Visual ratios are far below **1%**, we have:
>
> $$
> |\Delta\_{\text{cross}}| \ll \|e\_{\text{vision}}\|^{2} + \|e\_{\text{text}}\|^{2}.\tag{5}
> $$
>
> meaning that the interaction term contributes negligibly to the total error. This confirms that, under our quantization setting, cross-modal dependencies are weak and the additive form used in Eq. (5) is a well-justified first-order approximation.

---

> ### Author Response · Authors · 2025-11-30
> **Response to Reviewer hjdw (Part: 3/3)**
>
> **Q6: During iterative optimization, activation statistics change. Do you recompute $α$ at each iteration or fix it initially?**
>
> **Reply:** We appreciate the reviewer’s question. In our implementation, $α$ is computed once at the beginning and kept fixed throughout calibration. The value is estimated from a single forward pass over the calibration set, which reliably captures the inherent scale disparity between visual and textual activations at the layer level.
>
> We intentionally avoid recomputing $α$ during optimization, as repeated updates would introduce additional overhead and could destabilize the calibration process. In practice, we observe that this one-shot estimation remains stable and effective across all evaluated models.
>
> While dynamically updating $α$ is a potential direction for future exploration, our current results indicate that a fixed $α$ is sufficient to achieve strong and consistent performance.
>
> **Q7: FlatQuant assumes rotational invariance, but your modality weighting seems to break symmetry. Are these approaches compatible?**
>
> **Reply:** We appreciate the reviewer’s insightful question. To clarify upfront: **our modality weighting does not interfere with FlatQuant’s computation invariance**, and the two mechanisms remain fully compatible.
>
> The concern about “symmetry breaking” appears to stem from differing notions of invariance. In FlatQuant, rotational invariance is a *function-level* property: once the learned orthogonal rotation is folded into the weights, the forward computation and model outputs remain exactly the same. This invariance does not depend on the reconstruction loss being symmetric during calibration.
>
> Our modality weighting only reweights the reconstruction objective to adjust the optimization emphasis across modalities; it does not alter the rotation operator itself, its orthogonality, or its ability to be folded back into the model weights. As a result, FlatQuant’s invariance guarantee is fully preserved, and the weighting mechanism operates as a complementary calibration-level enhancement rather than a modification of the underlying rotation transformation.
>
> **Q8: Are rotation matrices learned jointly with quantization parameters? If so, how does modality weighting affect rotation learning?**
>
> **Reply:** We appreciate the reviewer’s question. **Yes, the rotation matrices are jointly optimized together with the quantization parameters** under the FlatQuant rotation–reconstruction framework, and are ultimately folded into the model weights to preserve computation invariance during inference.
>
> Our modality weighting does not alter the rotation operator itself. Instead, it modifies the reconstruction objective by rebalancing the relative contributions of visual and language tokens. This change affects the gradients used during calibration and thus indirectly guides the updates to the rotation matrices, encouraging them to better capture language-token characteristics. In this way, modality weighting serves as a lightweight objective-level refinement that complements the rotation learning process.

---

### Meta-Review · Area_Chair_d5W5 · 2026-01-06

**Summary:**

The reviewers acknowledged the paper's thorough experiments and clear responses, and raised many concerns. The authors provided strong, data-driven responses addressing most technical concerns (e.g., independence assumption, hyperparameter choice, ablation studies). There are also concerns about novelty and additional issues including insufficient analysis of computational overhead.

**Reviewer Concerns:**

The rebuttal well addressed concerns about the independence assumption, hyperparameter β sensitivity, and component ablation with clear experiments. However, outstanding issues remain: (1) whether performance gains primarily stem from FlatQuant rather than the proposed modality-aware weighting (novelty concern), (2) lack of concrete computational overhead metrics.

**Reviewer Scores:**

The reviewers did not participate in the discussion, so I think they may not modify the scores.
I appreciate the authors for the detailed rebuttal. However, the initial scores are low and no changes are made.

---

### Decision · Program_Chairs · 2026-01-26

Reject